# Design of H-Shape Chamber in Thermal Bubble Printer

**DOI:** 10.3390/mi13020194

**Published:** 2022-01-26

**Authors:** Xishun Peng, Anjiang Lu, Qiliang Sun, Naitao Xu, Yibo Xie, Jiawen Wu, Jin Cheng

**Affiliations:** 1College of Big Data and Information Engineering, Guizhou University, Guiyang 550000, China; pxs19970921@163.com; 2Wuxi V-Sensor Technology Co., Ltd., Wuxi 214000, China; abc810801@126.com (Q.S.); wuaven@outlook.com (J.W.); 3School of Optoelectronic Engineering, Xi’an Technological University, Xi’an 710000, China; xunaitao_123@163.com (N.X.); 13319215096@163.com (Y.X.)

**Keywords:** MEMS, thermal bubble, micro chamber, flow limiting structure

## Abstract

The utilization rate of ink liquid in the chamber is critical for the thermal bubble inkjet head. The difficult problem faced by the thermal bubble inkjet printing is how to maximize the use of ink in the chamber and increase the printing frequency. In this paper, by adding a flow restrictor and two narrow channels into the chamber, the H-shape flow-limiting structure is formed. At 1.8 μs, the speed of bubble expansion reaches the maximum, and after passing through the narrow channel, the maximum reverse flow rate of ink decreased by 25%. When the vapor bubble disappeared, the ink fills the nozzle slowly. At 20 μs, after passing through the narrow channel, the maximum flow rate of the ink increases by 39%. The inkjet printing frequency is 40 kHz, and the volume of the ink droplet is about 13.1 pL. The structure improves the frequency of thermal bubble inkjet printing and can maximize the use of liquid in the chamber, providing a reference for cell printing, 3D printing, bioprinting, and other fields.

## 1. Introduction

Since the debut of the first commercial inkjet printer, made by Hewlett-Packard Corporation, in 1984, the size of its market has been constantly expanding [1,2]. Up to now, the application field of inkjet printing is not only limited to daily office printing but also has become a hot topic in the fields of 3D printing and cell bioprinting [3,4]. At present, inkjet printers on the market are mainly divided into two kinds: thermal bubble inkjet printing and piezoelectric inkjet printing. Comparing with the latter, thermal bubble inkjet printing has a simple structure and lower cost. With the cell bioprinting field continuously heating in recent years, because of the unique working way of thermal bubble printing [5], it causes a little damage to cells. Therefore, the research on thermal bubble inkjet printing has become the hot topic again. The main reason is that the thermal bubble printing relies on heating resistors within 3 μs to instantaneously heat the ink to more than 300 °C, and the overall temperature of the whole chamber only rises about 5–10 °C [6,7], so most cells will not be inactivated by heating, in which the average survival rate is more than 90%. Piezoelectric inkjet printing can achieve an extremely high ejection frequency (up to 30 kHz) [8,9]. However, because the piezoelectric inkjet printing head is installed directly inside the printer, if the nozzle is blocked in the process of using it, it cannot be solved by replacing the ink cartridge, and the printer must be disassembled to replace the inkjet printing head, so its maintenance costs are high. The thermal inkjet printing head is connected to the ink cartridge for easy replacement. When the ink in the cartridge is used up, a new ink cartridge can be replaced. As a result of the low cost of the ink cartridge, there is no need to worry too much about the problem of nozzle blockage. Moreover, in the piezoelectric inkjet printing head, the sealing performance of the whole chamber is strictly required. Since the shape variable of the vibrating plate is very small, poor sealing will lead to a lot of deformation, which will seriously damage the cell membrane [10], and the extrusion force generated by the vibrating plate cannot be concentrated on the nozzle to effectively eject the cells, either. In addition, when the viscosity of the liquid in the chamber is too large, the vibrating plate will not be able to provide the corresponding shape variables, which results in the nozzle not working.

A lot of efforts have been made on the generation and ejection of ink droplet in thermal bubble inkjet printing. Salman Sohrabi et al. [11] proposed an inkjet nozzle with a diameter of 48 μm, which was used to simulate the inkjet process, and the heating conditions of the heating resistors are 3 μs–10 V–250 mA. In this structure, the power density of the heating resistors is small, and the power consumption is large. In addition, the entire inkjet period is relatively long. Hua Tan et al. [12] proposed an inkjet nozzle with a diameter of 20 μm to simulate the printing process. Due to the small size of the chamber, when the bubble expands, which will produce an explosive extrusion pressure to the liquid around it (not only in direction of the nozzle), some ink will be squeezed back into the main channel. Until the bubble liquefies and disappears, this part of the ink will flow back to the chamber. Then, because of the repeated high temperature and high pressure, the ink will appear to have chemical metamorphism, affecting the quality of inkjet printing. When applied to cell bioprinting, this phenomenon can also cause some cells to inactivate. For example, Allain et al. [13] proposed the use of a thermal printer (HP694C) for cell bioprinting, but the conventional structure (the nozzle is at the same height as the chamber, and the diameter of the nozzle is about 20 μm or 50 μm) is still defective, leading to some tissue fluids being repeatedly squeezed and heated, as shown in Figure 1, resulting in the cells inactivating.

## 2. Theoretical Model of H-Shape Flow-Limiting Structure

In this study, for the design of a thermal bubble inkjet chip, we propose to add a flow restrictor and two narrow rectangular channels, which forms an H-shaped flow-limiting structure. Figure 2 shows the three-dimensional structure of the inkjet printing head and the back structure of the chamber. The width of the flow restrictor is slightly less than the sum of the two heating resistors. The latter is nearly the same width as the diameter at the bottom of the nozzle, and its center is positioned directly in the direction of the nozzle, so as to maximize the extrusion force of the bubble, causing the ink droplets to eject out of the nozzle. The inkjet printing head designed in this paper has a diameter of 20 μm at the bottom of the nozzle, and the maximum width of the chamber is 64 μm. According to the bottom diameter of the nozzle, the reasonable width of the restrictor should be between 15 and 25 μm, and the width of the narrow channel on both sides should be set according to the expansion direction and the size of the bubble, and considering the backflow rate of the main channel. The reasonable width of the narrow channel should be about half of the width of the restrictor, that is, between 7.5 and 12.5 μm. The main purpose of this design is that when the bubble forms, the ink around it will be squeezed, which is squeezed out from the nozzle to eject and also squeezed back into the main channel. The flow restrictor prevents the bubble from expanding in the direction of the main channel, and it causes the bubble to release an extrusion force in the direction of the nozzle, so the ink in the chamber is fully utilized. When the bubble expands, the liquid reverse flow rate is fast. According to Newton’s inner friction theorem, the narrow channel has limited the flow rate. After the bubble began to liquefy, the liquid backflow rate is slow, and the narrow channel structure increases the inner surface of the wall and the contact area of the ink. At this time, the inner friction plays a guiding role to speed up the ink from the main channel to fill the chamber. If the width of the narrow channel is too large, the flow blocking effect cannot be achieved. However, if the width is too narrow, the backflow rate will be affected. The narrow channel forms by the flow restrictor and the chamber wall extends to the connection layer. Sufficient narrow channel length not only plays a role in blocking when the reverse flow rate is too large but also plays a role in guiding when the flow rate is too slow.

During the whole process in thermal bubble inject printing, in order to maximize the utilization rate of ink in the chamber, rather than making it reverse flow to the main channel, we need to understand the process of ink flowing in the chamber. First, we start by inferring the flow resistance in the nozzle. According to Poiseuille’s theorem [14,15], under the laminar movement, in steady state, incompressible flow, when there is liquid flow in a circular pipe, the volume of liquid flow *Q* has the following relationship with the radius *r* and length *l* of pipe, the pressure difference ∆*p* between the two ends of the pipe, and the viscosity coefficient ε of the fluid:(1)$$Q=\frac{\pi r^{4}\Delta p}{8\varepsilon l}$$

Actually, when we research fluid mechanics, the parameters of *r*, *l*, and ε are determined. In order to simplify the equation, we assumed the flow resistance coefficient *R =* 8ε*l/πr*^4^, thus:(2)$$Q=\frac{\Delta p}{R}$$

Since the thermal bubble nozzle is a frustum of a cone, for its cross-section that contacts with the chamber, its flow resistance coefficient $R_{c}$ can be inferred as:(3)$$R_{c}=\frac{8\pi \varepsilon l}{\pi^{2}r^{4}}=\frac{8\pi \varepsilon l}{S^{2}}$$
where *S* is the cross-section area of fluid, and the flow resistance coefficient of the rectangular pipe (the width *a* is twice the height *b*) is about:(4)$$R_{r}=\frac{321\varepsilon }{a^{2}b^{2}}=\frac{321\varepsilon }{S^{2}}$$
where *a* and *b* represent the width and height of the rectangular pipe, respectively. According to Equations (3) and (4), under the same condition of *S*, the flow resistance coefficient $R_{r}$ of the rectangular pipe is larger than that of the circular pipe, $R_{c}$.

## 3. Experimental Results

### 3.1. Simulation

In this paper, we use the finite element software to simulate the entire inkjet process. The thermal bubble inkjet chip designed by us is shown in Figure 2, which is composed of the main channel, connecting layer, chamber, and nozzle.

In this paper, as shown in Figure 3, we use the fluid dynamics model in the finite element software to conduct mesh division on the thermal bubble inkjet printing head, and a total of 159,633 grid cells are divided. In the simulation of thermal bubble inkjet printing, the key point is the driving conditions of a high-pressure vapor bubble. The model used in this paper simulates bubble formation by entering high-pressure vapor bubbles from below the heating resistors (‘R’ in Figure 2). We refer to the CFD model of Wilson et al. [16]; they did a lot of research on bubble expansion, but they lacked a real bubble model. Then, we also refer to Asai et al. [17]; the saturated vapor pressure inside the bubble will reach 10–100 times the atmospheric pressure. In their later studies [18,19], the initial bubble pressure was set at 7.5 Mpa. Therefore, the maximum bubble pressure during nucleation should be at least 7.5 Mpa. According to the actual physical debugging and matching of the chip, because of the large work density of the heating resistors designed in this paper, the bubble pressure here is set at 9 Mpa. In addition, we refer to the model used in reference [20], which drives the vapor bubble at high pressure and sets a liquid–vapor phase transition condition to simulate the bubble disappearance process. When the heating resistors stop pressurizing, the pressure inside the bubble decreases over time. A liquid–vapor transition condition set in this paper is that when it is equal to 1 atm (because the liquid is too shallow here, the pressure of water is not considered), the gas is converted to water, and finally, the bubble disappeared. In addition, the inner wall of the nozzle is set as the wetting wall, and the number of contact angles is 78 degrees. The back of plane G (‘G’ in Figure 2) is an open interface, where the conditions are set to inhibit reflow and the static pressure is 0. We do not consider the situation in which the ink is squeezed back to the cartridge by a bubble.

In order to obtain the most reasonable design size of the flow-limiting structure, firstly, the width of the flow restrictor needs to be inferred, and we need to determine the final size according to the flow rate. The equation of flow rate [21] is:(5)Q=S×V
where *Q* is the flow rate, *S* is the area of the flow cross-section, and *V* represents the flow velocity in the cross-section. As shown in Figure 4, at 1.8 μs, the width of the flow restrictor is 0, 5, 10, 15, 20, and 25 μm (the length is set to 10 μm), respectively, and the vertical axis represents the flow rate of planes A, B, and C respectively. A is the cross-section at the top of the nozzle, and B and C are the front and back cross-section of the narrow channel, respectively. In this paper, the area and position of plane A are fixed, while plane B and plane C will change with the different widths of the narrow channel.

Figure 4 shows the blocking effect on bubble expansion when the width of the flow restrictor is 0, 10, 20, and 25 μm, respectively. It can be found that when the width is 20 μm and 25 μm, the flow rate at planes B and C is less, and the flow rate at plane A is large. In fact, when the width is 20 μm, it is enough to block the bubble. Considering the machining accuracy and the flow rate when ink backflows from the main channel to the chamber, the width of 25 μm is too large and counterproductive. Therefore, we determine the width of the flow restrictor as 20 μm. It should be noted here that although the flow restrictor prevents the expansion direction of the bubble, because of its shorter length, it plays a poor flow-limiting role, and most of the ink liquid reverse flows to the main channel. According to the Newton inner friction theorem [22,23], the length of the pipe directly affects the flow resistance coefficient, so we extend its length to the connection layer, which increases the length of the narrow channel. The flow restrictor does not completely connect with the connecting layer, because the manufacturing process of the structure allows certain errors in the processing accuracy.

In Figure 5, at 1.8 μs, the narrow channel widths are 7, 9, 11, 13, 15, 17, 19, and 21 μm, respectively. The flow rate of planes A, B, and C is shown in Figure 5. When the width of the narrow channel is 7 μm, the flow rate at plane A is the maximum, and the flow rate at B and C is the minimum. However, we also need to consider the ink flow rate, after the bubble disappeared, from the main channel into the chamber. As shown in Figure 6, in order to study the effect of ink backflowing at different size narrow channels, we continue to study the flow rate at planes A, B, and C. The initial state is to set the ink to just fill the chamber and have no ink in the nozzle. At this time, the ink flow rate is 0 m^3^/s. After 3 μs, because of the capillary force, the liquid level in the nozzle began to rise, and we continue to study the flow rate at planes B and C. In this case, the flow rate of planes B and C is minimum when the width of the narrow channel is 7 μm. We should note that the change rate of the difference between *Q* at planes B and C is the largest when the width increases from 9 to 11 μm. At this time, the flow-blocking effect of the narrow channel is obvious. Considering the ejection flow rate and the obstruction of bubble expansion, the width of the narrow channel is determined to be 11 μm. The design parameters of the inkjet printing head structure are shown in Table 1.

In this paper, we selected several important moments to display the initial state of the vapor bubble, including the top view when the bubble expands to the maximum, the ink droplet separates from the ink liquid, the bubble shrinks, the bubble disappears, the ink liquid surface in the nozzle rises, and finally, the concave surface is formed after filling. Figure 7a represents the initial state of thermal bubble inkjet printing. We simulate the formation of bubbles by entering high-pressure steam bubbles with an initial pressure at 9 atm. After the driving signal is applied, the surfaces of the two heating resistors begin to nucleate a thin layer of vapor bubbles. At this moment, the bubbles are small, and there is a certain distance between the two bubbles.

As shown in Figure 7b, when the driving signal stops working after 1.8 μs, the bubble expansion velocity reaches the maximum value. Since bubble expansion creates extrusion pressure, the surrounding ink is squeezed and flows around. The bubbles continue to expand for some time due to the inertia of the fluid. It should be noted that during the period when the bubble continues to expand, its expansion velocity is positive and its acceleration is negative.

Figure 7c shows that the bubble expands to the maximum at 2.5 μs; we have analyzed the reason why bubbles continue to expand after 1.8 μs. It can be seen from Figure 7b that the droplets above the nozzle are thick at the top and thin at the bottom, which is caused by Newton’s inner friction at the nozzle and the viscous force between the ink liquid. When the flow rate is slow, the inner friction generated on the wall of the pipe has the guiding function of flow; when the flow rate is too fast, the force on the wall of the pipe acts to obstruct the flow. The principle of viscous force [24,25] of fluid is as follows:(6)$$\tau =\varepsilon \frac{dv}{dy}$$
where *τ* is the viscous force, *ε* is the viscosity coefficient, and $\frac{dv}{dy}$ is the speed gradient in the normal direction of the plane. The fluid between the two plates can be divided into countless fluid layers parallel to the plate, and there are speed differences between layers. As a result of the attraction between fluid molecules, the faster layer of fluid drags the slower layer forward. At the same time, the same number of molecules in the slower flow layer enter the faster layer, exerting an equal and opposite decelerating force on it. This transfer continues layer by layer until it reaches the wall, eventually creating friction. At the same time, we can see the top view at 2.5 μs from Figure 7c. As the bubbles expand, two small bubbles on the heating resistors fuse into a large bubble and squeeze the surrounding ink in the chamber. Since the end of the chamber is closed, the main expansion direction of the bubble is at the nozzle, and there is reverse extrusion toward the main channel. The flow restrictor is positioned in the middle of the chamber to prevent a vapor bubble from expanding to the main channel, and two rectangular channels are formed between the flow restrictor and the wall of chamber. According to Equation (4) inferred above, the flow resistance coefficient of the rectangular pipe is not only related to the cross-section area *S* and fluid viscosity coefficient ε but also to the length *l* of the pipe. Therefore, we extend the length of the rectangular channel to the connection layer to prevent ink reverse flowing to the main channel. Actually, in the field of cell bioprinting, the biggest harm of thermal bubble inkjet printing is that the tissue fluid in the chamber is repeatedly heated at high temperature. This is because the bubble expansion squeezes part of the tissue fluid reverse flows to the main channel, and then, the bubble liquefies and shrinks, and the tissue fluid backflows to the chamber. In this process, due to the repeated heating, some cells have not been ejected and lose their activeness, thus reducing the overall survival rate of the cell group in the tissue fluid. In fact, whether it is ink or tissue fluid, repeated heating will reduce the printing quality. The H-shape structure not only solves this problem, but also, because of the maximum utilization of ink in the chamber, it improves the overall inkjet printing frequency.

Figure 7d has shown that the ink droplet separates from the ink liquid at 4.5 μs; this phenomenon is due to the driving signal having stopped working, and the ink droplet continues to move forward, while the viscous force between the fluids is not enough to drive the slow ink to continue to eject. At this time, the liquid at the nozzle shrinks inward and the surface tension is greater than the viscous force of the high-speed fluid on the low-speed fluid. Finally, a fracture is formed between the liquid ink, the ink droplet continues to eject, and the residual ink shrinks to the nozzle mouth. It should be noted that the bubble begins to collapse and become smaller after 2.5 μs reaches its maximum. At this time, the pressure of the outer layer of the vapor bubble becomes smaller; when it is equal to 1 atm, the gas–liquid transition condition is reached, and the bubble is liquefied to ink. This process will attract ink from the nozzle and the main channel to fill the bubble.

Figure 7e has shown the flight path of an ink droplet in the air domain at 6 μs. Here, in order to reduce the complexity of the model, we set the length of the air domain to 60 μm. It can be seen from Figure 7e that the ink droplet has almost flown out of the air domain, and the bubble further becomes smaller, while the height of the liquid level in the nozzle also decreases at the same time. When the time comes to 9 μs, we can see from Figure 7f that the vapor bubble in the chamber has completely liquefied, and the liquid level in the nozzle finally drops to the bottom. It should be noted that the height of the liquid level at this time is higher than 20 μm of the chamber thickness. Therefore, this will not cause air to be sucked back into the chamber, resulting in bubble accumulation, and finally reducing the printing quality due to insufficient ink supply in the nozzle.

It can be seen from Figure 7g that when the time comes to 23 μs, the ink level in the nozzle rises, almost filling the entire interior. Actually, the reason why we chose to put Figure 7g in this paper is that when the bubble disappears, the ink level inside the nozzle will rise because of the effect of capillary force; then, the two sides of the ink level to reach at the outlet of the nozzle wall first and then stop rising, and the central surface presents a concave shape. At this time, if the amplitude of the concave shape needs to be further reduced, it needs the force of surface tension and gravity. The contraction of surface tension makes the liquid level as much as possible in the same horizontal plane. In addition, because the nozzle is inverted in practice, there is gravity, which further stretches the concave surface. When the internal friction force, surface tension, and gravity are in dynamic balance, the concave surface in the middle of the liquid surface has a small amplitude and tends to be flat, as shown in Figure 7h, and the process has been slow. In fact, the volume of the nozzle is about 8 pL (pL = 10^−12^ L = 10^−15^ m^3^), and the inkjet printing head (nozzle, chamber, and connecting layer are included) is about 97 PL. The volume of the ink droplet ejects each time, which accounts for about 10–15% of the inkjet printing head. In order to speed up the inkjet printing frequency, the complete filling of the nozzle can be ignored in actual printing, and the second time, the inkjet can be started when the two sides of the liquid surface are close to the end of the nozzle.

In order to better understand the inkjet state of the droplet, planes B and C were selected to analyze the flow rate. At 1.8 μs, the maximum flow rate at plane B is about 3.2 × 10^−9^ m^3^/s, and the maximum flow rate at point C is about 2.4 × 10^−9^ m^3^/s. After passing through the narrow channel, due to the influence of friction, the ink flow rate is reduced by 25%. In order to further observe the flow direction of the ink under the flow-limiting structure, we continue to analyze the cross-section of the whole structure. As shown in Figure 8a, the section in the middle of the chamber is selected for fluid analysis. It can be found that at 1.8 μs, the bubble expansion speed reaches its maximum, and the flow speed in the narrow channel is between 10 and 14 m/s. The existence of the flow-limiting structure prevents most ink from reverse flow and ensures the maximum utilization of ink in the chamber. In fact, the size of the flow restrictor and narrow channel directly affects the flow speed of liquid, and we need to consider the effect of the flow-limiting structure on liquid backflow filling after the bubble disappeared. As shown in Figure 7f, at 9 μs, the bubble disappeared completely, and the ink liquid in the nozzle will be supplemented by capillary force. The liquid flow speed in the narrow channel is shown in Figure 8b. At this time, the ink liquid in the main channel fills the chamber through the narrow channel, and the maximum flow speed in the narrow channel is about 1.2 m/s.

Then, we continue to analyze the flow rate of the ink liquid in the nozzle. The plane A is the cross-section at the top of the nozzle. As we can see from Figure 9a, the maximum inkjet flow rate at plane A is about 18.41 × 10^−9^ m^3^/s, and the maximum inkjet speed at the center of the plane A is about 18 m/s. The nozzle diameter in Ref. [11] is 46 μm, and because the work density of the heating resistors is low, the droplet speed at the nozzle is 12 m/s. The structure proposed in this paper, due to the flow-limiting effect of the flow restrictor and narrow channel, further increases the bubble extrusion pressure in the nozzle direction, and the heating resistors have the large work density, so its ejection speed is greater than 12 m/s. The nozzle diameter in Ref. [12] is 20 μm, and the inkjet speed is about 26 m/s. This is because the heating time of its heating resistor is 3 μs, while in this paper, it is only 1.8 μs. However, the inkjet period in Ref. [12] is 50 μs, which is much larger than the structure designed in this paper.

As shown in Figure 9b, through the narrow channel, the ink liquid flows into the chamber from the main channel, and the flow rate at plane B is faster than that at plane C, speeding up the filling rate of ink in the nozzle. At about 20 μs, the maximum reverse flow rate is about 0.31 × 10^−9^ m^3^/s at point C. When the ink liquid passes through the narrow channel, the maximum flow rate of plane B is about 0.43 × 10^−9^ m^3^/s. It can be found that after the guidance of a narrow channel, the ink flow rate in the back and front cross-sections increased by nearly 39%. Later, due to the nozzle diameter shrinkage and other factors, the backflow rate begins to slow down, and at 30 μs, the whole nozzle is almost filled to reach the conditions for the second inkjet printing.

### 3.2. Device Fabrication

In this paper, the School of Optoelectronic Engineering, Xi’an Technological University was commissioned to use MEMS (micro-electro-mechanical system) micromachining technology to manufacture thermal bubble inkjet chips. Using a bare silicon wafer as the substrate and TaN (tantalum nitride) as the heating resistors, a SU-8 photoresist preparation chamber and jet nozzle were prepared above it. The specific flow diagram is shown in Figure 10. First of all, a piece of bare silicon is prepared as a substrate (a), and a layer of Si_3_N_4_ (silicon nitride) dielectric layer is deposited on the substrate to prevent the electric current from entering the silicon (b). Then, a layer of TaN and Al (aluminum) is deposited as the main device of the heating resistor (c), and the heating resistors and wire are prepared by mask, exposure, and etching (d). The TaN here serves as the heating resistors, and the current inflows and outflows through Al above it, and the electric current runs through the TaN and heats it up. Later, a Si_3_N_4_ dielectric layer is then prepared above the heating resistors to prevent current from flowing into the chamber and causing the electrolytic of the ink inside it. After the heating resistors are made, a layer of 20 μm SU-8 photoresist is pasted above it, and the flow channel and chamber are prepared by exposure, development, and baking processes (e). Finally, another layer of 20 μm SU-8 photoresist is attached above the first layer of the SU-8 photoresist, and the nozzle is prepared by exposure, development, baking, and other processes again (f).

As shown in Figure 11a, in this paper, the length of the thermal bubble inkjet chip is designed to be about 15 mm, which integrates 360 inkjet printing heads. Figure 11b shows the internal structure of the inkjet chip. It can be found that the inkjet printing heads are evenly placed on both sides of the main channel. The square in the chamber is the flow resistor, with two narrow rectangular channels to form the H-shape structure, when the bubble expands, to prevent ink reverse flow in the main channel. It also ensures that the extrusion pressure generated by each bubble’s expansion is maximized in the direction of the nozzle. Therefore, it ensures that as much of the ink liquid in the chamber as possible is ejected from the nozzle.

### 3.3. Thermal Bubble Inkjet Printing Test

For inkjet printing, the two most critical parameters are the volume of the droplet and the printing frequency. The former determines the printing accuracy, and the smaller the ink droplet, the higher the printing accuracy. The latter determines the printing time. When the diameter of the nozzle is reduced and the ink droplet becomes smaller, the ink marks line will become thinner and thinner. At this time, it is necessary to increase the printing frequency and eject more ink droplets to ensure the width of the ink marks line. According to the simulation results, we get the H-shaped structure of the thermal bubble inkjet printing head, and the whole inkjet period is controlled within 30 μs. In fact, in the previous simulation results, we mentioned that at 23 μs, the liquid level inside the nozzle is almost full enough to support the second time inkjet printing. Therefore, in the following experiment, we control the inkjet period at 25 μs and the thermal bubble inkjet frequency up to 40 kHz. The extra 2 μs serves as a buffer time to ensure that the amount of ink in the nozzle is sufficient and the volume of ink droplets ejected out is basically unchanged under the high-frequency continuous inkjet printing. When the ink droplets from the nozzle hit the glossy photo paper of the electric turntable, they form an ink marks line. If the distance between two adjacent ink droplets is consistent, the size of the ink droplets is uniform and the surface is smooth. Then, it shows that the inkjet period of the H-shaped inkjet print head designed in this paper is 25 μs, and the printing working frequency is 40 kHz, which is consistent with the previous simulation results.

In the preparation process of the test, as shown in Figure 12a, we fixed the height of the inkjet printing box, which was about 1 cm away from the electric turntable. The reason is that the ink droplet is too small, and it is very prone to be disturbed by the micro air flow in the air during the falling process, making its flight path deviate. The electric turntable is attached to a ring of glossy photo paper, instead of ordinary wooden paper, because after ink droplets are printed on wooden paper, there will be infiltration and diffusion, which is not conducive to later observation, while glossy photo paper can lock ink droplets well to avoid their changes, which is convenient for later observation. The electric turntable is driven by an electric motor at 500 RPM (rounds per minute), and the period of the motor *T* is about $\frac{3}{25}$ seconds per round. Next, we first start the electric turntable; the driving conditions of the electric motor are 24 V–500 mA. Next, we choose one of the inkjet printing heads and set the inkjet printing times to 800. As shown in Figure 12b, the driving conditions of the heating resistors are 1.8 μs–12 V–120 mA. The period of inkjet printing is 25 μs. Then, we get an ink marks line. The distance $r_{t}$ between each point on the curve to the center of the turntable is 3.8 cm.

Using the optical microscope to observe the ink marks line, as shown in Figure 13, there are 12 ink droplets randomly selected for analysis. Due to the high rotation speed of the turntable, when the ink droplets hit the paper, they will drift due to their own flow characteristics, so the ink droplets on the paper were not round. Next, we continue to observe the edges of the droplets and find the centers of the 12 droplets before they drift. The marked center lines show that they are almost on the same horizontal line, not a curve. The reason is that the distance between two adjacent droplets is too small. According to Newton’s calculus theorem, the length of the curved line can be broken down into the accumulation of innumerable linear lines, while the distance between two adjacent droplets is about 50 μm. Therefore, the linear velocity $V_{t}$ corresponding to the turntable at the ink droplet location can be set, which approximates the velocity of a straight line under constant motion, and speed $V_{t}$ can be calculated by the following formula:(7)$$V_{t}=\frac{2\pi r_{t}}{T}.$$

In the above equation, $r_{t}$ = 3.8 cm, and the period of the motor *T* is about $\frac{3}{25}$ seconds per round, so the distance $L_{t}$ between adjacent ink droplets is:(8)$$L_{t}=V_{t}\times t\approx 49.7\times 10^{-6}(m).$$

Then, in Figure 13, from left to right, we need to measure the distance between two adjacent points of 12 ink droplets, which adds up to 11 groups. As shown in Figure 14, the vertical axis represents the distance between two droplets, and the horizontal axis represents the sequence number from the first group to the last.

An auxiliary line is given in Figure 14, corresponding to the distance of 49.7 × 10^−6^ (m). It can be found that the measured distances of the 11 groups are basically consistent with the theoretical experimental results. It should be noted here that because the diameter of the ink droplet is about 20 μm and the size is small, there will be some errors in the accuracy of the measuring distance and the selection of mark points in the center of the ink droplets. Therefore, the distances of the 11 groups in Figure 14 do not completely coincide with the theoretical results, which is reasonable.

Next, we need to calculate the volume of each inkjet droplet. As mentioned above, the volume of the inkjet printing head designed in this paper is 97 pL. If the volume of the inkjet droplet ejects each time, which accounts for about 10–15% of the whole inkjet printing head, then it meets the standard. As shown in Figure 15a, because the volume of the ink droplet ejected each time is too small to be calculated, we use one inkjet printing head to continuously eject 200,000 times and use a glass slide to undertake the droplets. We choose the glass slide instead of glossy photo paper because it has a smooth surface and is chemically stable, making it almost impossible to be penetrated by ink droplets. At the end of the ejection, a hemispherical droplet of ink forms on the surface of the glass slide. Then, we use an endoscope to observe it and adjusting the angle of the endoscope to be parallel to the desktop. As shown in Figure 15b, the proportionality coefficient of the width to height of the hemispherical droplet can be obtained, which is about 7:1. As shown in Figure 15c, in order to get the specific size of its width, we use the optical microscope to measure it. The cross-section of the ink droplet is round, and there is an aperture in the center of the ink droplet, which is the reflection of the fill light of the microscope. Finally, we get the width of 3.57 mm and the height of 0.51 mm.

Next, calculating the volume of the hemispherical droplet. As shown in Figure 16, given that the high $H_{t}$ of the sphere is 0.51 mm and the radius $r_{t}$ of the sphere is 1.785 mm, the Pythagorean theorem can be used to solve the radius $R_{t}$ of the sphere:(9)$$R_{t}{}^{2}={\left(R_{t}-H_{t}\right)}^{2}+r_{t}{}^{2}.$$

By solving Equation (9), it can be obtained that $R_{t}$ is about 3.379 mm. Next, we infer the volume $V_{t}$ of the hemispherical droplet, and the solution process is as follows:(10)$$x^{2}+y^{2}=R_{t}{}^{2}$$
(11)$$V_{t}=\int_{R_{t}-H_{t}}^{R_{t}}\pi x^{2}dy=\pi \int_{R_{t}-H_{t}}^{R_{t}}\left(R_{t}{}^{2}-y^{2}\right)dy=\pi H_{t}{}^{2}(R_{t}-\frac{H_{t}}{3}).$$

By solving the equations above, the volume $V_{t}$ of the hemispherical droplet is approximately 2.62 × 10^−9^ m^3^ (2.62 × 10^−6^ L); since the total of 200,000 droplets are ejected, the volume of each ink droplet is approximately 13.1 pL. This experimental result is consistent with theoretical estimation.

## 4. Conclusions

In this paper, the H-shape flow-limiting structure is proposed and verified by simulation. The top diameter of the nozzle designed in this paper is 18 μm, the working time of the heating resistors is 1.8 μs, the time from bubble expansion to disappearance is controlled within 9 μs, and the whole inkjet period is controlled within 25 μs. We analyzed the flow restrictors with widths of 0, 5, 10, 15, 20, and 25 μm respectively, and we found that the optimal size was 20 μm. Then, we continued to analyze the narrow channel with the widths of 7, 9, 11, 13, 15, 17, 19, and 21 μm, respectively. In this process, the flow-limiting effect and the backflow effect should be considered. Finally, the width of the narrow channel is determined to be 11 μm. During the simulation experiment, the maximum limiting flow rate at both ends of the narrow channel is about 25%, and the maximum guiding flow rate is about 39%. In the inkjet printing test, an electric turntable (500 RPM) is used to test the frequency of the inkjet printing head, and we chose one inkjet head for continuously ejecting 800 times. Under the optical microscope, it can be clearly seen that the distance between adjacent ink droplets basically consists with the theoretical distance of 49.7 × 10^−6^ (m). Then, an inkjet printing head is used for continuously ejecting 200,000 times, and the glass slide is used to undertake ink droplets. The volume of the hemispherical droplet is obtained by the Pythagorean theorem and calculus, and the volume of each ink droplet is finally calculated, 13.1 pL, which reaches the expected goal. The existing mainstream inkjet printing heads, HP, and Canon’s thermal bubble inkjet frequency is mostly at 15 kHz, while the inkjet printing head designed in this paper can reach 40 kHz, even higher than some piezoelectric inkjet printing heads. In addition, due to the flow-limiting effect of the H-shaped chamber, the inkjet volume is about 13.1 pL, and the ink utilization rate of the chamber is up to 13.5%. The H-shape structure not only speeds up the thermal bubble inkjet printing frequency but also, due to its flow-limiting effect, provides a new design structure for inkjet equipment in cell printing, 3D printing, bioprinting, and other fields.

## Figures and Tables

**Figure 1 micromachines-13-00194-f001:**
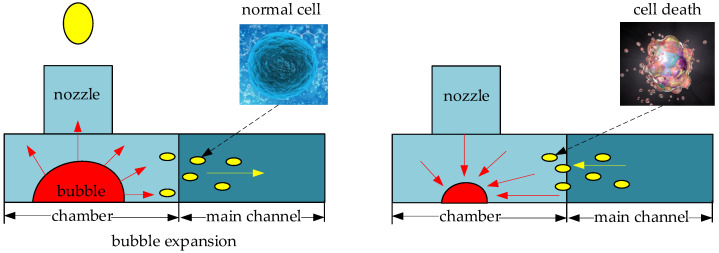
Cells were repeatedly heated to inactivate.

**Figure 2 micromachines-13-00194-f002:**
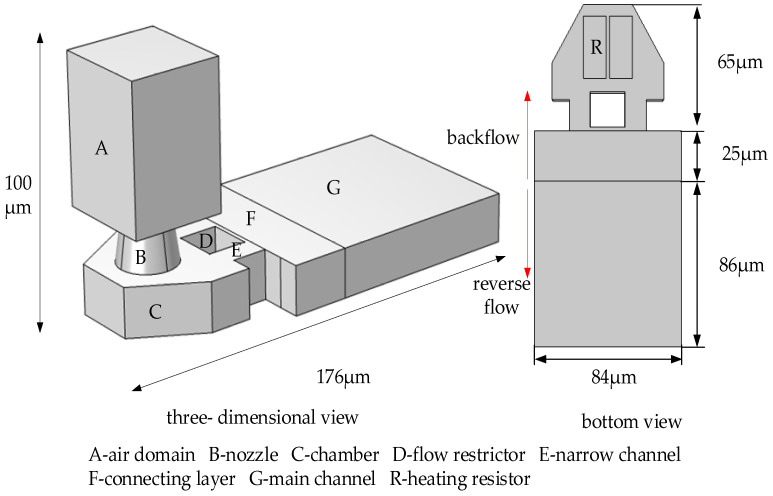
The three-dimensional structure of the inkjet printing head and the back structure of the chamber.

**Figure 3 micromachines-13-00194-f003:**
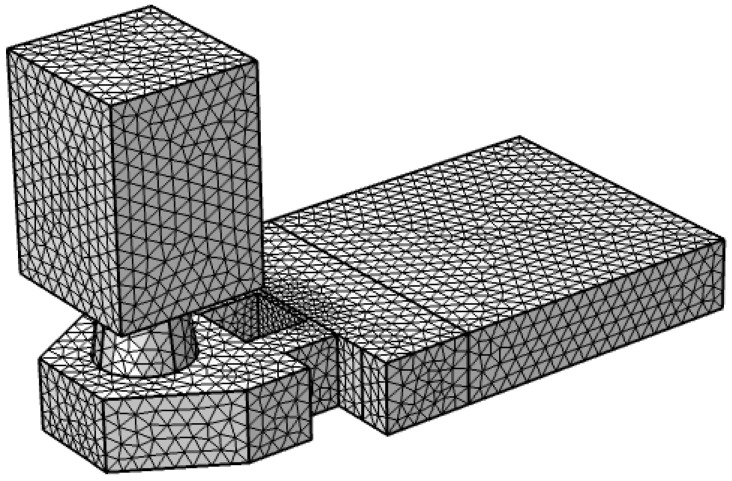
The thermal bubble inkjet printing head was meshed by using fluid dynamics in finite element software.

**Figure 4 micromachines-13-00194-f004:**
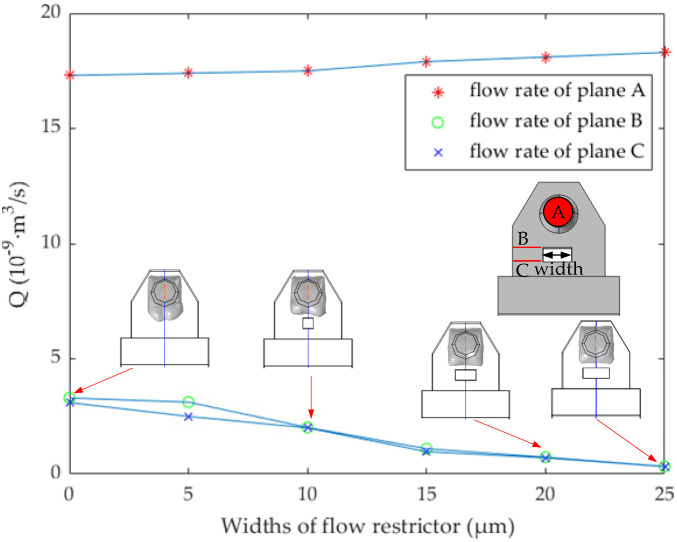
At 1.8 μs, the flow rate of planes A, B, and C at different widths of the flow restrictor.

**Figure 5 micromachines-13-00194-f005:**
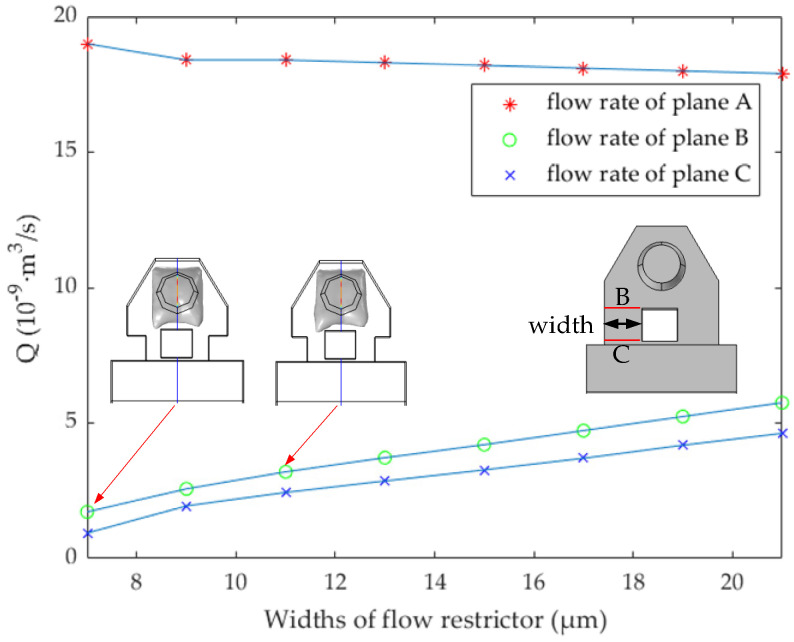
At 1.8 μs, the flow rate of planes A, B, and C at different widths of the narrow channel.

**Figure 6 micromachines-13-00194-f006:**
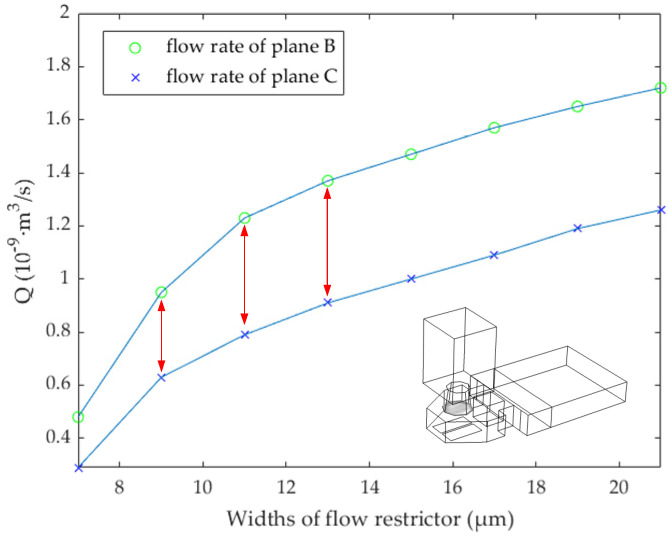
At 3 μs, the backflow rate of planes B and C at different widths of the narrow channel.

**Figure 7 micromachines-13-00194-f007:**
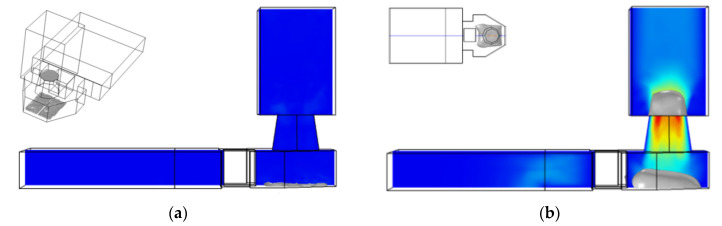
The process of bubble inkjet printing. (**a**) The initial state of inkjet printing. (**b**) Inkjet printing at 1.8 μs. (**c**) Inkjet printing at 2.5 μs. (**d**) Inkjet printing at 4.5 μs. (**e**) Inkjet printing at 6 μs. (**f**) Inkjet printing at 9 μs. (**g**) Inkjet printing at 23 μs. (**h**) Inkjet printing at 30 μs.

**Figure 8 micromachines-13-00194-f008:**
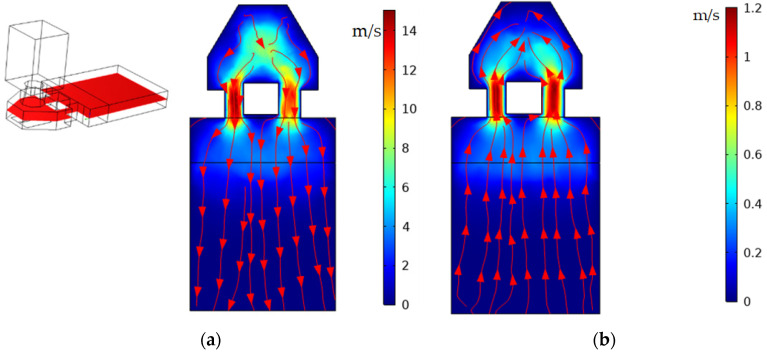
Analyzing the flow speed of an H-shaped structure. (**a**) Analyzing the flow speed of a plane at 1.8 μs. (**b**) Analyzing the flow speed of a plane at 9 μs.

**Figure 9 micromachines-13-00194-f009:**
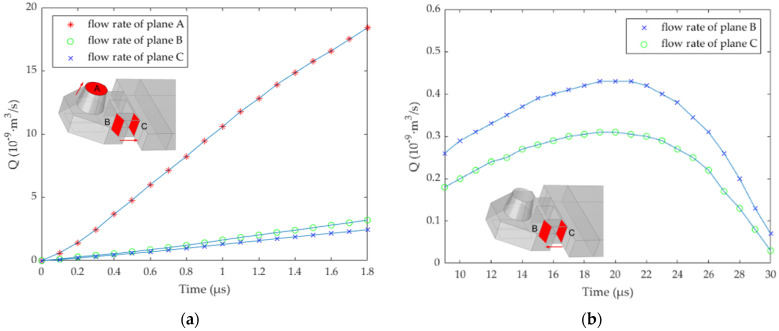
Analyzing the flow rate of the H-shaped structure. (**a**) Analyzing the flow rate of planes within 1.8 μs. (**b**) Analyzing the flow rate of planes during backflow.

**Figure 10 micromachines-13-00194-f010:**
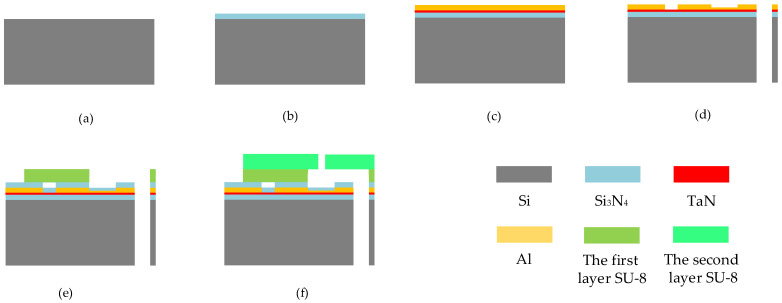
Manufacturing process in the thermal printing chip. (**a**) Using a bare silicon wafer as the substrate. (**b**) Using the Si_3_N_4_ as the thin layer. (**c**) Using the TaN and Al to make heating resistors. (**d**) Etching the chip by MEMS process. (**e**) Using the first layer of SU-8 photoresist to make chamber. (**f**) Using the second layer of SU-8 photoresist to make nozzle.

**Figure 11 micromachines-13-00194-f011:**
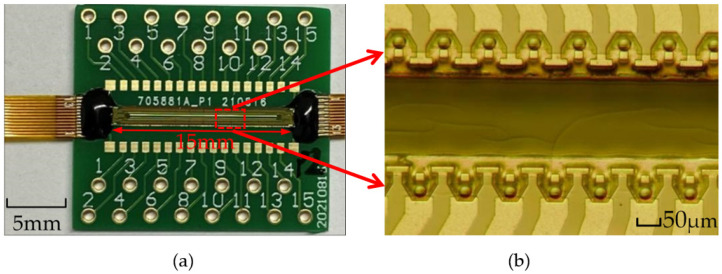
Thermal bubble inkjet chip with H-shape structure. (**a**) An overall preview of the thermal bubble inkjet chip. (**b**) Internal structure of the thermal bubble inkjet chip.

**Figure 12 micromachines-13-00194-f012:**
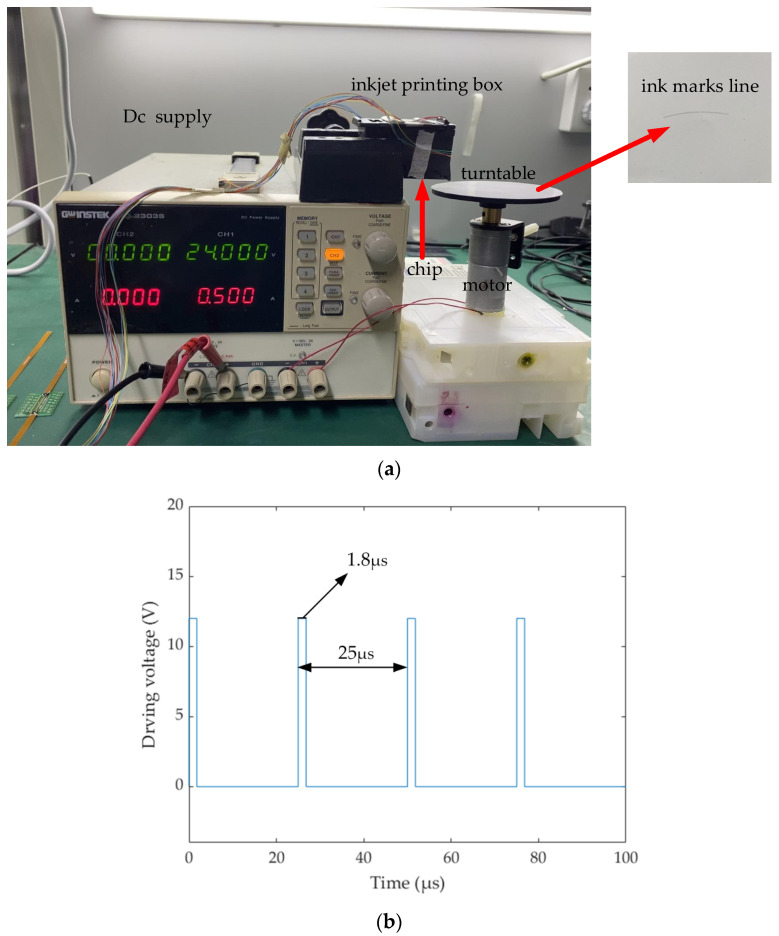
Preparation for inkjet experiment. (**a**) Using the electric turntable to measure frequency. (**b**) The drive signal of heating resistors.

**Figure 13 micromachines-13-00194-f013:**
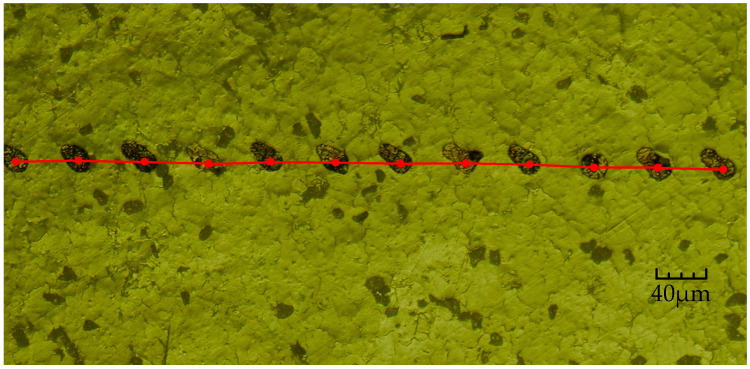
Under the optical microscope (made by Shenzhen Aosvi Optical Instrument Co., Ltd., Shenzhen, China), the ink marks line on an electric turntable.

**Figure 14 micromachines-13-00194-f014:**
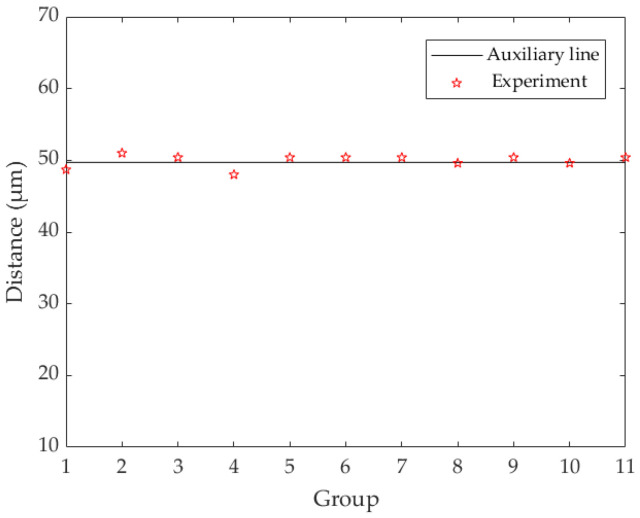
From left to right, the distance between two adjacent droplets.

**Figure 15 micromachines-13-00194-f015:**
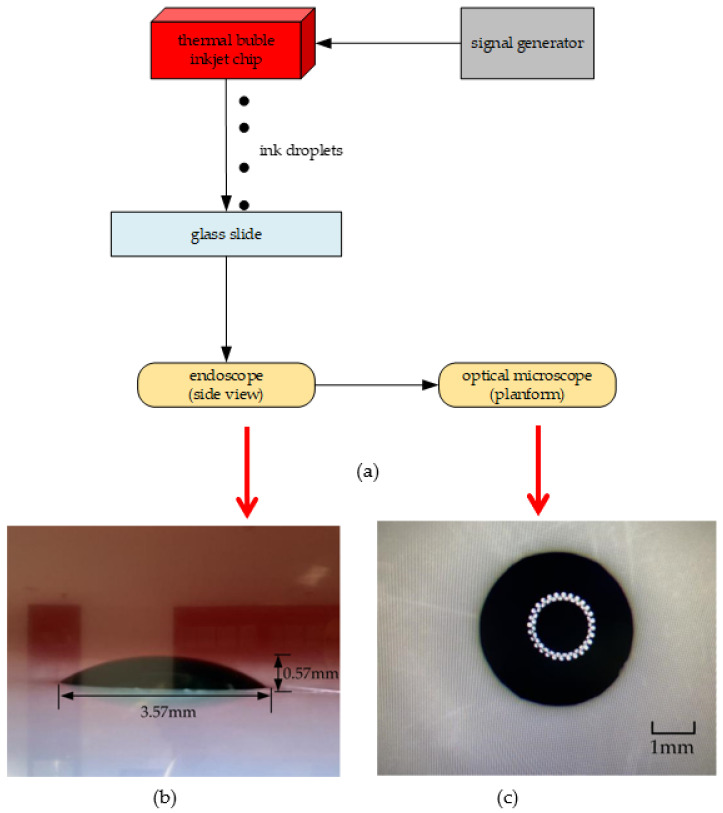
Measurement of the volume of the hemispherical droplet. (**a**) Continuously ejecting of ink droplets 200,000 times. (**b**) Using the endoscope to measure the proportionality coefficient of width to height of the hemispherical droplet. (**c**) Using the microscope to measure the width of the hemispherical droplet.

**Figure 16 micromachines-13-00194-f016:**
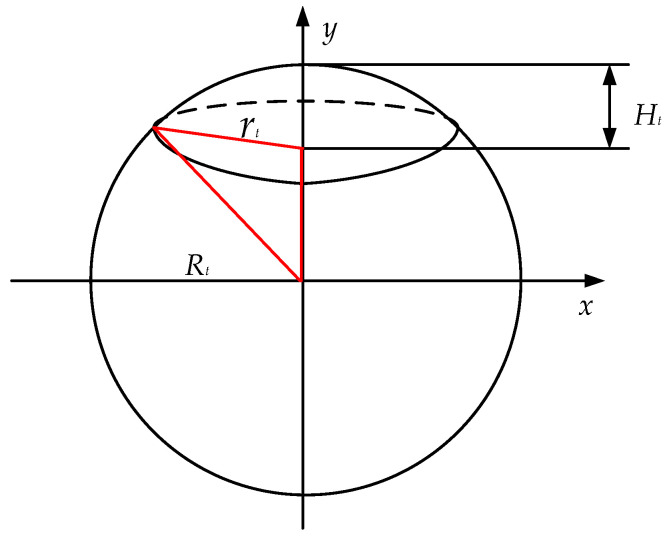
Derivation of volume of hemispherical droplet.

**Table 1 micromachines-13-00194-t001:** The design parameters of the inkjet printing head.

Structural Parameters	Value (μm)
Nozzle top diameter	18
Nozzle bottom diameter	20
The nozzle height	20
Heating resistor	13 × 32
Main channel	84 × 91
Connecting layer	84 × 20
Flow restrictor	20 × 18
Narrow channel	11 × 14
Chamber thickness	20

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
