# Peer review of "Design of H-Shape Chamber in Thermal Bubble Printer"

_micromachines, 2022, doi:10.3390/mi13020194_

Round 1

Reviewer 1 Report

This paper describes H-shape chamber in thermal bubble printer to maximize the use of ink and increase the printing frequency. The topic of this review would be interesting to the readers.
However, some of my concerns are:

1. In Fig. 1, is this the conventional method or from reference?
2. In Fig. 2, I believe this is the proposed design, and especially D and E seems being important. Even though the text mention the dimension, however, it would be nice to mention dimension range.
3.  In section "2. Theoretical model of H-shape flow limiting structure", authors discussed theoretical models. Were these equations such as eq.(14) used in the later section or for explain the concept of the design etc.
4. In the equation (13), authors described flow resistance coefficient for rectangular pipe (or for parallel plate?). I think another paper[1] refers the combination of width and length also contributes to the flow resistance. Will there be restrictions to apply equation (13) to general rectangular pipe.  
[1]H. Bruus, Lecture Notes in Theoretical Microfluidics, Department of Micro and Nanotechnology, Technical University of Denmark, 3rd edn, 2006, p. 31.
5. In fig. 9, it would be good to mention the mesh size to explain any non-symmetry streamline if it is based on FEM.
6. How this study is compared with reported study in terms of printing frequency and the volume of ink droplet?

Overall, authors proposed a new geometrical design as a solution with validations from theoretical and experimental aspect. The reports should be attractive to the readers. However, I believe some details and explanation are missing, which are my concerns.

Author Response

The revised manuscript has been uploaded, and the modified descriptions are attached at the end of the article,  p.g.19 to 22.

Reviewer 2 Report

The authors carried out a fluidic design of printhead for thermal inkjet (TIJ) printing using numerical simulations and fabricated the TIJ device based on the design for testing. The manuscript has some novelty and is relevant to the journal. The paper is not well written and organized. The English language needs significant improvement. It seems that the authors do not understand the fluid physics of TIJ device very well, so some interpretation of simulation results is not correct. I recommend major revisions before consideration for publication in the journal. My specific comments are

(1) In Section 1, the authors mentioned that piezoelectric inkjet printing (PIJ) is easy to produce satellite droplets and has difficulty in handling high viscosity liquids. These comments are not correct. Both PIJ and TIJ can produce satellite droplets with improper firing conditions. The breakup of liquid filament into multiple droplets depends on fluid properties and operation conditions. PIJ is well known to handle high viscous fluid better than TIJ. I recommend the authors rewrite the comparison between TIJ and PIJ.

(2) Section 2 should be removed or significantly shortened, as the laminar flow in pipes of different cross-section is well established. Eq. 14 can not be used to obtain the height of meniscus after refilling, because the the capillary force induced by the meniscus in the nozzle is balanced with the back pressure ( vacuum pressure at the inflow) of the printhead instead of the gravity. Without back pressure, the ink will overflow the orifice plate.

(3)As the simulation is a key in the work, the authors should present their CFD model with details such as input parameters, mesh sensitive study, boundary conditions, validation, etc. For simulation of TIJ process, modeling of the vapor behavior in the explosive boiling process is critical. The authors should introduce how they implement the drive bubble in their CFD model.

(4)In Section 3.1, the authors used the velocity at different positions (A,B,C in Figs. 5-7 and 9) to evaluate the flow through the nozzle and reverse flow during the droplet ejection process. It is not proper, as the select of the locations is quite arbitrary and can not represent the overall trend in the flow well. A better way is to calculate the instant flow rate through a plane (e.g, a plane that cuts through the nozzle, another plane that cuts through the narrow channels). The authors should use the flow rate instead of velocity in the discussion.

(5) Discussions related to Fig. 8 were poorly presented. The initial growth of the vapor bubble is driven by the high initial bubble pressure from boiling event. As the vapor bubble grows rapidly, the bubble pressure drops significantly (e.g. less than atmospheric pressure after a few microseconds) . So the bubble expands much more slowly after initial growth phase. The reason that the vapor bubble can still grow even after the resistor is turned off is due to fluid inertial.  I strongly recommend the authors study more references related to TIJ process. The authors need to improve the Section 3.1 significantly.

(6) The authors made a mistake of unit in Line 314 on p.g. 10. pL=pico-liter=1x10^-12L.  The drop volume in Line 508 on p.g. 15 was estimated to be 2.62x10^-9L (or 2.62 nanoliter). This seems too small. For a droplet on millimeter scale, the volume should be on the order of micro-liter (10^-6L). As a result, the authors estimated their inkjet droplet volume to be 13.1x10^-15L (line 509 on p.g. 16). The number does not look right. For a nozzle of diameter on tens of micrometer, the droplet volume is on the order of pico-liter (10^-12L). I highly doubt these volume numbers presented throughout the paper.

Author Response

The revised manuscript has been uploaded, and the modified descriptions are attached at the end of the article,  p.g.19 to 23.

Round 2

Reviewer 1 Report

Thank you for your detailed explanation. Though, the explanation was not clear for point 4. Probably, I did not explain it thoroughly.

Q1: Is equation (4) good to study both rectangle channels and square channels, and is this equation good under steady state, incompressible flow or some other condition.   

In steady state, incompressible flow, some flow resistances I saw in some microfluidic papers/lectures are
Rrectangular ~ 12/(1-0.63(h/w) * mu*L * 1/(h^3w) (when w>h)
Rsquare ~ 28.4*mu*L*1/w^4
The equations used some approximations when solving second derivatives, and have certain errors depending on w/h selection. These equations differ from what authors shown in the eq. (4). That's why I thought I was missing some information. My understanding is: from Fig. 3 in original manuscript, authors derived the hydraulic resistance of rectangle channels from Newton's law of viscosity relation or planar Couette flow. The Newton's law of viscosity relation is generally used to study the liquid velocity between two parallel plates. In the original manuscript, authors extended Newton's law of viscosity relation from parallel plates to rectangle channels. While many studies use stokes equation, which has second derivatives, to derive the hydraulic resistance of rectangle channels. I apologize if I misunderstood your equation. Please correct me if I am wrong.

Q2:mesh settings at narrow neck E looks a bit coarse. Have you tried any fine mesh. or How long did it take to complete one simulation.

LL 141, 142. CFD model of reference[16], which does a nice job in simulating bubble expansion.  
-> ref [17]?

Other than above points, the paper has greatly improved.

Author Response

Please see the manuscript and cover letter.

Reviewer 2 Report

I'm satisfied with the authors' revisions in terms of technical aspects. The English of the manuscript still needs significant improvement before acceptance for publication in the journal.

Author Response

Dear reviewer,

please see manuscript and cover letter for revision instructions. 

Round 3

Reviewer 1 Report

Thank you for your detailed explanation. I am satisfied with authors' explanation. As authors mentioned the derivations were done decades ago, I believe it would be better to mention/cite reference(s).